# A Hybrid Learning Framework for Predicting Post-Treatment Serum Sodium in Patients with Hyponatremia

1st Ghanahshyam B. Kshirsagar,[1]
Member, IEEE
College of Engineering,
The Roux Institute, Northeastern
University, Portland, ME, USA
g.kshirsagar.ece@northeastern.edu

1st Robert Hayden[1]
Division of Nephrology,
Department of Medicine
Maine Medical Center
Portland, ME, USA
robert.hayden@tufts.edu

2nd Prasheen Shah
Division of Nephrology,
Department of Medicine
Maine Medical Center
Portland, ME, USA
prasheen.shah@mainehealth.org

3rd Christopher El Mouhayyar
Division of Nephrology,
Department of Medicine,
Massachusetts General Hospital and
Harvard Medical School, Boston, USA
celmouhayyar@mgh.harvard.edu

4th Sahir Kalim
Division of Nephrology,
Department of Medicine,
Massachusetts General Hospital and
Harvard Medical School, Boston, USA
skalim@mgh.harvard.edu

5th Sagar Nigwekar
Division of Nephrology,
Department of Medicine,
Massachusetts General Hospital and
Harvard Medical School, Boston, USA
snigwekar@mgh.harvard.edu

6th Raimond Winslow
College of Engineering,
The Roux Institute, Northeastern
University, Portland, ME, USA
r.winslow.ece@northeastern.edu

7th Qingchu Jin*
College of Engineering,
The Roux Institute, Northeastern
University, Portland, ME, USA
q.jin.ece@northeastern.edu

*Abstract*—Hyponatremia (serum sodium [S-Na$^+$] <135 mEq/L) is common in hospitalized patients and linked to adverse outcomes. Management of severe hyponatremia ([S-Na$^+$] <120 mEq/L) remains controversial and challenging, requiring a controlled rate of correction—typically a rise in [S-Na$^+$] of 6 mEq/L over 24 hours—to reduce the risk of complications in high-risk patients. Rapid correction (>8 mEq/L/day) has been associated with a higher risk of developing osmotic demyelination syndrome, while overly slow correction has been associated with longer hospital stays. Achieving the target correction often requires 3% hypertonic saline (3% NaCl) and sometimes desmopressin acetate (DDAVP). The required amount of 3% NaCl varies with patient-specific factors and clinician judgment, often resulting in suboptimal correction due to the complexity of predicting individual responses.

Predictive modeling of post-treatment [S-Na$^+$] may offer a promising solution. While mechanistic models may simulate fluid-electrolyte dynamics, and machine learning (ML) models may offer additional data-driven insights, existing models are limited by non-individualized predictions and small, homogeneous datasets, thus limiting their generalizability.

To address these gaps, we developed a hybrid modeling framework that integrates mechanistic physiology with ML-based prediction. We developed our approach using two internal cohorts drawn from Massachusetts General Hospital and Brigham and Women's Hospital. Cohort 1 (n=144) included heterogeneous-risk patients, and Cohort 2 (n=73) was a high-risk subset of DDAVP-treated patients from Cohort 1 (primary data), which was then externally validated on MIMIC-IV. Our hybrid model outperformed existing models, particularly in the high-risk Cohort 2. We submit that this hybrid modeling framework can be applied to other clinical challenges with small and complex clinical datasets.

*Keywords*—*Hyponatremia, Predictive Modeling, Serum Sodium Correction, Physiological Equations, Machine Learning*

## I. INTRODUCTION AND LITERATURE

Hyponatremia, a common electrolyte disorder encountered in clinical practice, is defined as a [S-Na$^+$] concentration below 135 mEq/L [1]. In most cases, hyponatremia reflects an excess of total body water (TBW) in relation to total body sodium (TB-Na$^+$), which can lead to a variety of symptoms, including nausea, vomiting, fatigue, as well as serious complications such as cerebral edema, seizures, and coma in cases of severe hyponatremia [2, 3]. An increase in [S-Na$^+$] concentrations can be achieved either by an increase in (TB-Na$^+$), a decrease in TBW, or a combination of both.

Managing severe hyponatremia remains a challenging and controversial clinical task. Many authorities on managing severe hyponatremia recommend a correction rate corresponding to an increase of 6 mEq/L in [S-Na+] over 24 hours to reduce the risk of adverse outcomes associated with both rapid and slower correction [4, 5]. The treatment strategy to achieve this target correction goal in many cases includes the use of intravenous (IV) 3% NaCl with frequent monitoring of [S-Na$^+$] levels to ensure appropriate correction [6-8]. Specific subsets of patients at high risk of fast correction can also have a marked increase in free water loss from urine output after initial treatment with 3% NaCl, leading to an exaggerated and unpredictable correction rate. IV DDAVP (a synthetic antidiuretic hormone that acts on the kidney to increase water reabsorption) may be used in such cases to reduce the rate of free water excretion through urine output, thereby slowing the overall correction rate and lowering the risk of rapid correction [9, 10]. The amount of 3% NaCl given to achieve the desired correction rate in each case varies considerably depending on a wide variety of variables, including individual patient characteristics, real-time data on urine output, and clinician experience [11, 12]. Given this variety of individual factors, prediction of [S-Na$^+$] in response to any given treatment plan is difficult and often

Ghanahshyam Kshirsagar and Robert Hayden contributed equally.
*Corresponding Author

results in suboptimal correction rates associated with worse outcomes [4-13]. Retrospective studies have shown that 28% - 42% of patients experienced fast correction (defined as at least > 8 mEq/L increase in 24 hours) [11-13] and 29% experienced slower correction (defined as < 4 mEq/L in 24 hours) [13].

While several physiological equations have been proposed to predict sodium correction [$\Delta Na^+$] rates, including the Edelman [14], Rose [15], and Adrogué-Madias (A-M) [16] equations, these models typically assume steady-state conditions and can be sensitive to missing or imprecise clinical variables. These equations [14-16] may also not capture individual variability in water balance, osmotically inactive compartments, or real-time data from electronic medical records (EMRs), which ultimately limits their accuracy and clinical utility in long-term [$S-Na^+$] prediction.

The widespread implementation of electronic health record (EHR) systems has resulted in vast patient data repositories, enabling the application of data-driven clinical decision support tools to enhance hyponatremia management [17-22] potentially. There is emerging interest in the initial development of ML models that can better predict change in [$S-Na^+$] concentrations post-treatment [17]. For example, a ML model developed using a small cohort of hyponatremic patients accurately predicted [$S-Na^+$] levels based on only a few routinely collected clinical variables [18]. In another investigation, a pilot study evaluating ML approaches for forecasting the occurrence of postoperative hyponatremia reported strong performance metrics, with an area under the receiver operating characteristic curve (AUROC) of 84.3% and an overall accuracy of 78.4% [19]. In a recent study, a support vector regression (SVR) model predicted [$S-Na^+$] concentrations using 7 clinical features—including water intake, sodium [$Na^+$] intake, potassium [$K^+$] intake, urine volume, [$S-Na^+$], potassium, and chloride ($Cl^-$) levels—with high accuracy ($R^2 = 0.92$) and low root mean square error [20]. However, the study was conducted on a small cohort of only 15 subjects, indicating high uncertainty in its findings and potential overfitting of the data. Furthermore, it excluded broader patient populations and high-risk subgroups such as those treated with DDAVP or presenting with low urinary sodium [$U-Na^+$]. This narrower focus may limit utility in real-world, dynamic clinical settings with heterogeneous patient profiles. Moreover, despite the accuracy for [$S-Na^+$], these models often underestimate the actual [$\Delta Na^+$], namely the change of [$S-Na^+$] before and after treatment, and do not explicitly predict [$\Delta Na^+$], which limits their clinical reliability and acceptance.

Furthermore, modeling hyponatremia using retrospective EHR data may be challenging due to high data missingness (e.g. often exceeding 50–60% for critical variables in publicly available databases like the Medical Information Mart for Intensive Care IV (MIMIC-IV) [21] and electronic Intensive Care Unit Collaborative Research Database (eICU) [22]), delayed documentation, and irregular sampling, all of which degrade data fidelity and impair both physiological and ML models. Mechanistic models [14-16] may show high error rates when key inputs like total body water, urine electrolytes, or fluid balance are imprecise or missing, while ML models [17-20] tend to overfit and generalize poorly due to sparse

data, latent confounding (e.g. unmeasured predictors), and unbalanced etiologies.

To address these challenges, we obtained a high-quality dataset extracted by manual chart review from clinical nephrologists and hypothesized that a hybrid modeling approach combining physiological constraints with data-driven learning can better handle missing and inaccurate inputs, reduce prediction errors, and improve robustness. Furthermore, we specifically included a patient population presumably at higher risk of fast correction, as determined by the treating clinicians: patients treated with DDAVP. To achieve this objective, we implemented the proposed hybrid learning method across two distinct cohorts: Cohort 1, which served as the primary cohort, encompassed patients who received DDAVP as well as those who did not; Cohort 2 specifically focused on those patients who had received DDAVP. The experiments involving Cohort 1, consisting of 144 patients with 294 treatment intervals (referred to as intervals in this study), which is ten times greater than the study by Kinoshita et al. [20], and Cohort 2, comprising 73 patients with 149 intervals, utilized a combined dataset of de-identified patient records obtained from two separate research institutions. Additionally, the MIMIC-IV database [21] was utilized as a secondary validation resource for our model's ability to predict post-treatment [$S-Na^+$] and [$\Delta Na^+$]. Our hybrid model demonstrated a statistically significant improvement over the mechanistic and existing machine learning models within the high-risk (DDAVP-treated) cohort. In this work, we introduce several key findings for predicting [$S-Na^+$] and [$\Delta Na^+$]:

1. The main contribution is a more precise model derived from a hybrid modeling pipeline, which outperforms the mechanistic model.
2. Our study addresses key limitations of the existing studies by including a high-quality dataset extracted by manual chart review from clinical nephrologists and by including a high-risk DDAVP-treated cohort
3. We introduced a new performance metric, binary error rate (BER), that accounts for measurement uncertainty in [$\Delta Na^+$] modeling.
   We conducted a sub-cohort-specific evaluation in high-risk patients receiving DDAVP, confirming that the hybrid model delivers more stable correction trajectories and significantly reduces prediction variability in controlled [$Na^+$] management protocols.

## II. DATA

### A. Primary data

We identified a cohort of patients from Massachusetts General Hospital (MGH) and Brigham and Women's Hospital (BWH) with severe hyponatremia ([$S-Na^+$] ≤ 120mEq/L). A total of 3,100 patients met the following initial inclusion criteria: 1) age ≥18 years, 2) at least one ICU admission at MGH or BWH between January 2016 and January 2024, 3) at least one [$S-Na^+$] level of ≤130 mEq/L (this initial [$S-Na^+$] threshold was later lowered to ≤120 mEq/L), and 4) at least one measurement of urine sodium [$U-Na^+$] and urinary potassium [$U-K^+$]. From this, 2,842 patients were excluded for one or more reasons: absence of severe hyponatremia ([$S-Na^+$] ≤ 120mEq/L) within 24 hours of the

candidate interval period of time, absence of concurrent hyperglycemia (glucose >250 mg/dL), or concurrent renal impairment (serum creatinine $\geq$ 1.5 mg/dL). Additionally, nine patients were excluded due to a lack of urine electrolyte measurements coinciding with [S-Na$^+$] values. After applying these criteria, 249 patients with 480 valid intervals remained, where intervals are defined as the duration of patient stay between two consecutive [S-Na$^+$] measurements. All clinical data were meticulously obtained and adjudicated by independent nephrologists performing manual chart reviews. To ensure high temporal resolution and data completeness, we filtered intervals to include those where a) the interval duration was less than 12 hours, b) all urine electrolytes (Na$^+$ and K$^+$) were available, and c) complete interval fluid input and accurate urine output data were reliably recorded, as verified by manual chart review.

The resulting dataset comprised 150 patients with a median age of 67 years (interquartile range: 57–76), contributing a total of 311 post-treatment [S-Na$^+$] measurements. The details of the patients' characteristics are in Table I.

Table I: The details of the patients' characteristics

| Category | Characteristic | Value |
|---|---|---|
| Demographics | Total patients[a] | 150 |
| | Median age (IQR) | 67 (57–76) |
| | Patients aged $\geq$80 years | 44% |
| | Male patients | 66 (44%) |
| Clinical Intervals | Total clinical intervals | 311 |
| | Intervals with DDAVP administration | 158 (50.1%) |
| Contributing Etiologies[b] | Hypovolemia | 80 (53.3%) |
| | SIADH[c] | 74 (49.3%) |
| | Polydipsia | 25 (16.7%) |
| | Low Solute Intake | 32 (21.3%) |
| | Other | 15 (10.0%) |
| Medical History & Comorbidities | Prior hyponatremia | 26 (17.3%) |
| | Concurrent malignancy | 46 (30.7%) |

[a] One unique patient had two separate ICU admissions and both separate encounters were included
[b] More than one etiology may contribute to the hyponatremia
[c] Syndrome of inappropriate antidiuretic hormone secretion

Each interval was documented using 65 features, as presented in Table S1 (Supplement). Out of 150 patients, 6 exhibited paradoxical clinical patterns that were physiologically unexpected based on the documented interval inputs and outputs and were labeled as impossible cases. Impossible cases were identified under two situations: 1) an interval increase in [S-Na$^+$] despite both net loss of interval solute and concurrent net gain in interval free water, a combination in which [S-Na$^+$] should theoretically remain stable or decrease; 2) an interval decrease in [S-Na$^+$] following both a net interval loss of free water and a net interval gain in solute. These contradictions suggest possible documentation errors, timing mismatches, or inaccuracies in recorded fluid inputs. Because these cases could not be reliably interpreted based on the available clinical data, they were excluded to preserve the integrity and physiological validity of the analysis. The final dataset comprised 144 patients with 294 intervals. This larger group is identified as our primary cohort, Cohort 1. We then identified a smaller subgroup within Cohort 1, comprising only patients who received DDAVP, which we categorized as Cohort 2, consisting of 73 patients and 149 intervals. We refer to Cohort 2 as the high-risk group, since

the administration of DDAVP is an additional therapeutic strategy that many clinicians utilize for a subset of patients with hyponatremia who are at high risk of fast correction in their [S-Na$^+$]. Our study protocol and data usage were approved by the Institutional Review Boards (IRBs) of Massachusetts General Brigham Institutional Review Board and Northeastern University Institutional Review Board. The need for informed consent was waived.

*B. Secondary Data-External Validation*

We also used the MIMIC-IV database [21] as a secondary validation source for our model's ability to predict post-treatment [S-Na$^+$] and [$\Delta$Na$^+$]. MIMIC-IV, a publicly accessible, detailed clinical database created by Massachusetts Institute of Technology (MIT) and sourced from Beth Israel Deaconess Medical Center, includes electronic health records for over 250,000 patients and more than 500,000 hospital admissions between 2008 and 2019. It features hospital-wide and ICU data, including demographics, vital signs, lab results, medications, diagnoses, procedures, and clinical outcomes. Using the same inclusion and exclusion criteria (see Figure S1) as in our primary dataset, we identified patients with severe hyponatremia (serum sodium <120 mmol/L), resulting in 47 patients with 145 treatment intervals in Cohort 1. Of these, 36 patients with 65 intervals comprised the high-risk DDAVP-treated group (Cohort 2).

## III. METHODS

*A. Hybrid Model Framework*

Fig. 1 illustrates the architecture of a hybrid predictive framework that merges a mechanistic physiological model with a data-driven ML model. The mechanistic model calculates the post-treatment [S-Na$^+$] concentration by utilizing core physiological parameters, including initial [S-Na$^+$], estimated total body water (TBW), net solute change ($\Delta$Solute) which includes urine electrolytes, and net water change ($\Delta$TBW). Simultaneously, the ML model is trained on a feature set comprising 65 structured clinical variables, including demographic details, etiological classifications, current clinical conditions, comorbidities, medication use, fluid input/output dynamics, and temporal laboratory markers. See Table S1 for more information.

To integrate the outputs, a late fusion ensemble strategy is employed, where the final predicted [S-Na$^+$] is determined as the average of predictions from both the mechanistic and ML models. This averaging method facilitates the combination of the mechanistic model's physiological consistency with the ML model's capacity to identify nonlinear patterns and higher-order interactions within the data, ultimately enhancing generalization and robustness in varied clinical environments.

The Edelman equation serves as the primary model for plasma [$Na^+$] concentration [14], which Rose et al. subsequently modified it to calculate post-treatment [S-Na$^+$] by [15]. The mathematical representations of the rearranged Rose equation are as follows:

$$[S-Na^+] = \frac{(TBOsm + \Delta Solute)}{TBW + (\Delta TBW)} / 2 \qquad (1)$$

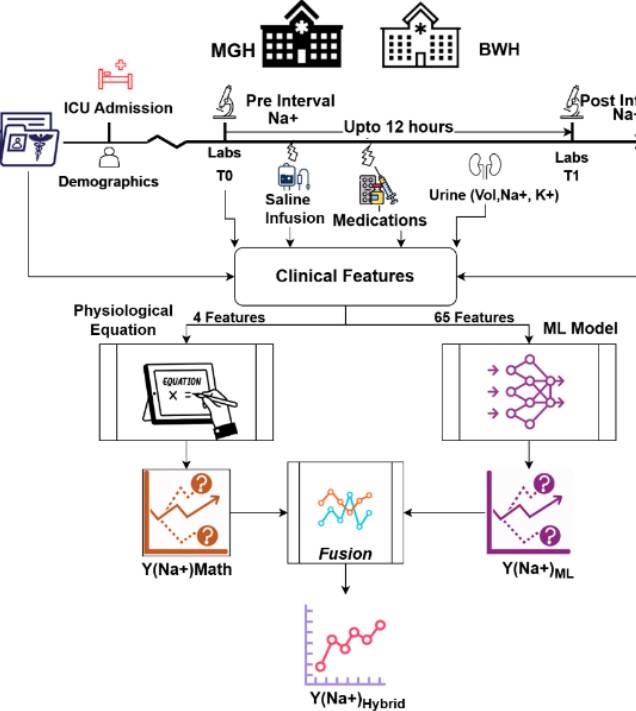

**Fig. 1**: Hybrid Predictive Framework Integrating a Mechanistic and ML Model for [S-Na⁺] Prediction.

where, $TBOsm$ stands for total body osmoles (total body solute contributing to overall osmolarity of body fluids, as further defined in Eq. 2), ΔSolute stands for net interval solute change (note that multiplying by 2 accounts for the anions that must accompany the positively charged Na and K; further defined in Eq. 3), ΔTBW stands for net interval water change (further defined in Eq. 4)

$$TBOsm = [Na^+] * 2 * TBW \qquad (2)$$

$$\Delta\text{Solute} = \Delta(Na + K) * 2 = [\,Solute_{In} - Solute_{out}\,] \quad (3)$$

$$Solute_{In} = [0.154 * 0.45\% \, NaCl] + [0.308 * 0.9\% \, NaCl] \\ + [1.026 * 3\% \, NaCl] + [0.260 * LR]$$
$$Solute_{out} = [Urine \, [Na +] + Urine \, [K +]] * 2 * \\ [Urine\_volume * 0.001]$$
$$\Delta\text{TBW} = \text{Water}_{In} - \text{Water}_{out} \qquad (4)$$

$$Water_{In} = [\text{Dextrose 5\%}] + [PO] + [LR] + [\text{Manual\_Water\_In}] + [3\% \\ NaCl] + [0.9\% \, NaCl] + [0.45\% \, NaCl]$$

$$Water\_out = Urine\_volume$$

where LR is the Lactated Ringer's solution, an isotonic crystalloid solution that closely resembles the composition of human plasma, PO refers to interval oral fluid intake, and Manual_Water_In accounts for any other interval water intake.

In the data-driven aspect of our hybrid framework, we utilized an ensemble of mechanistic and one of the best-performing of cutting-edge regression algorithms: Lasso Regression [23], Support Vector Regression (SVR) [20, 24], Extreme Gradient Boosting Regressor (XGBoost) [25], and TabNet [26]. The choice was based on their complementary strengths in managing high-dimensional, nonlinear, and diverse data. Lasso Regression employs L1 regularization,

allowing for simultaneous feature selection and coefficient shrinkage, which is especially beneficial in our case with 65 clinical predictors. SVR has shown strong generalization in earlier research concerning [S-Na⁺] prediction tasks, particularly in situations with limited and noisy data. XGBoost, a tree-based ensemble method that utilizes gradient boosting, excels in capturing intricate, nonlinear interactions and has demonstrated top-tier performance across a range of biomedical regression tasks. TabNet was chosen due to its demonstrated effectiveness with small-to-medium-sized tabular medical datasets, which offers a good balance between interpretability and reducing overfitting, and it outperforms traditional transformer models.

The final prediction $\hat{y}$ is derived through the late fusion of the predictions of the mechanistic and best-performing ML model:

$$\hat{y} = \tfrac{1}{2}(y_{Eq} + y_{ML}) \qquad (5)$$

Where, $y_{ML}$ is predicted [S-Na⁺] from the best-performing ML model, $y_{Eq}$, is the predicted [S-Na⁺] from Eq. (1).

This post-prediction fusion enables the model to leverage the benefits of both physiological modeling and data-driven prediction, thereby enhancing generalizability and robustness across diverse patient profiles.

*B. Model Development Pipeline and Performance Metrics*

We developed prediction models for [S-Na+] and ΔNa+ through a two-phase process. Initially, we utilized all 65 features (refer to Table S1) to implement mechanistic machine learning models, including Lasso, SVR, XGBoost, TabNet, and hybrid techniques, with internal validation. Subsequently, these models were retrained using a subset of 21 shared features between our primary dataset and the MIMIC-IV database (see Table S2 in the supplement), with evaluations conducted under both internal and external validations. This methodology was devised to assess performance under ideal conditions with a complete feature set and under constrained conditions with a limited variable set. A significant challenge with MIMIC-IV was the missing data for essential predictors such as total body osmoles (TBOsm), urine volume, and urine electrolytes (Na+, K+). To address this, we employed a KNN-based imputation strategy, setting K = 3 [27].

All models were trained and validated through five-fold cross-validation, with hyperparameters fine-tuned via grid search. We assessed performance using the mean squared error (MSE), BER (%), and the coefficient of determination (R²), averaging across folds to ensure robust and generalizable [S-Na⁺] predictions for clinically relevant subgroups.

This study introduces a new performance metric, the BER (%), which incorporates both the measurement uncertainty inherent in [S-Na⁺] determination (typically SD ≤ 1 -1.5 mEq/L for compliant laboratories [35]) and the established clinically accepted error for [ΔNa⁺][4]-[5].

To calculate the BER (%), we define a prediction as correct if the absolute difference between the predicted and actual [ΔNa⁺] is less than 1 mEq/L, given the fact that the [S-Na⁺] experimental uncertainty is in the range of 1mEq/L, and incorrect otherwise:

$$Prediction = \begin{cases} If \, |\Delta Na^+ - \widehat{\Delta Na^+}| \leq 1 mEq/L, correct \\ Otherwise, \, Incorrect \end{cases} \quad (7)$$

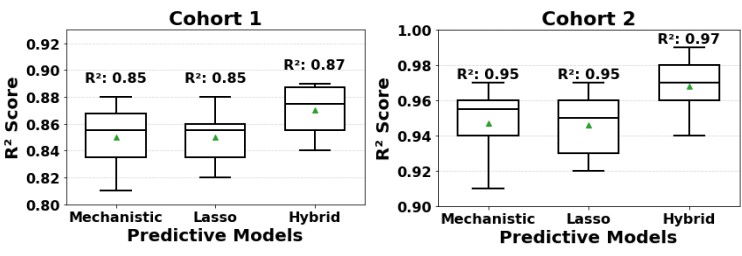

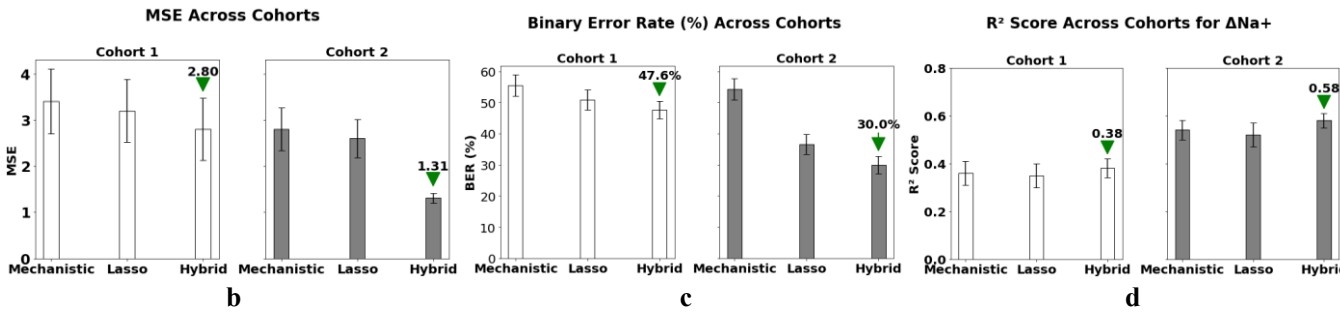

**Fig. 2**: Results of Internal validation of our Primary data with mechanistic model, Lasso and hybrid model for predicting post-treatment [S-Na$^+$] and [ΔNa$^+$] across two cohorts: (a) R$^2$ distribution over 5-fold cross-validation, (b) Mean squared error, (c) Binary error rate < ±1 mEq/L, and d) R$^2$ Score Across Cohorts for [ΔNa+]. Green arrows in (b) and (c) highlight the lowest error. The error bar represents the standard error.

Where, $\widehat{\Delta Na^+}$ - Predicted sodium Correction, and $\Delta Na^+$ is the actual sodium correction

$$BER\ (\%) = \frac{\#\ Total\ Predictions\ -\#\ Correct\ Predictions}{\#\ Total\ Predictions} * 100 \quad (8)$$

[ΔNa$^+$] is computed as:

$$\Delta Na^+ = Na^+_{predicted\ at\ T(n)} - Na^+_{basline\ at\ T(n-1)} \quad (9)$$

Where, $Na^+_{predicted}$ is the predicted post-treatment [S-Na$^+$], and $Na^+_{baseline}$ is pre-treatment [S-Na$^+$], and T(n) is interval time between the $Na^+_{predicted}$ and $Na^+_{baseline}$.

Non-parametric statistical tests, including the Wilcoxon signed-rank test [28] and the McNemar's Chi-square test [29], were applied to compare the performance across models. For the Wilcoxon signed-rank test, the squared error metric was used to compare models. McNemar's Chi-square test utilized a 2×2 contingency table of binary correct/incorrect classifications to compute the p-value based on differences in BER. These metrics were selected as primary evaluation criteria because they provide a more robust assessment of model performance than $R^2$ in the context of our study.

### C. Model interpretability

We investigated the contribution of 65 clinical features (Table S1, Supplement) to the prediction of [S-Na$^+$] levels in the hybrid and Lasso models, aiming to enhance interpretability. We employed SHAP (SHapley Additive exPlanations) [30], which assigns contribution scores to each feature by considering its impact across all possible feature combinations, ensuring fair attribution. Since our hybrid model is an average ensemble of a physiological equation and Lasso regression, we computed SHAP values separately for each component on the unseen test set from one representative fold of the 5-fold cross-validation. The absolute SHAP values were averaged across samples within each model and then combined by averaging to obtain final feature importance scores that reflect the

ensemble contribution. SHAP summary plots were used to visualize the top 8 features most influencing [S-Na$^+$] predictions.

## IV. RESULTS

This section presents the performance evaluation of the post-treatment [S-Na$^+$] and the correction from the baseline [ΔNa$^+$] using physiological equations, ML, and the proposed hybrid approach. Additionally, it includes SHAP-based feature interpretation.

### A. Internal validation (with all 65 features): Prediction of Post-treatment [S-Na$^+$] and [ΔNa$^+$]

Fig. 2 and Table II summarize the predictive performance of mechanistic, ML/DL (Lasso, XGBoost, SVR, and TabNet), and hybrid models in estimating post-treatment [S-Na$^+$] and [ΔNa$^+$] across two cohorts: Cohort 1 (primary) and Cohort 2 (high-risk patients treated with DDAVP). The MSE was used to assess prediction accuracy, while R$^2$ quantified the variance explained, and BER provided a clinically grounded measure of prediction reliability.

In Cohort 1, the hybrid model achieved the lowest MSE for [S-Na$^+$] prediction (2.8) (see Fig. 2b), improving over the mechanistic model (3.4) and Lasso regression (3.2). At the same time, XGBoost (5.37), SVR (5.34), and TabNet (4.24) exhibited significantly higher errors. In the high-risk Cohort 2, all models showed improved accuracy under controlled correction, with the hybrid model again achieving the best performance (MSE = 1.31) (see Fig. 2b), followed by Lasso (2.6), mechanistic (2.8), XGBoost (3.7), SVR (4.26), and TabNet (4.06). The R$^2$ values align with these findings: in Cohort 1, the hybrid model achieved an R$^2$ of 0.87 (See Fig. 2a), outperforming mechanistic (0.85), Lasso (0.85), XGBoost, SVR (0.83), and TabNet (0.84). Cohort 2 showed a similar pattern, with the hybrid model attaining the highest R$^2$ (0.97) (see Fig. 2a), exceeding the mechanistic and Lasso models (~0.95), while XGBoost, TabNet, and SVR trailed with R$^2$ values of 0.92, 0.91, and 0.90, respectively.

To evaluate the model's ability to predict physiologically meaningful [S-Na$^+$] changes, we conducted a post-hoc analysis of predicted [ΔNa$^+$], defined as the difference between predicted post-treatment [S-Na$^+$] and baseline pre-treatment [S-Na$^+$]. In Cohort 1, the hybrid model again outperformed all baselines with the lowest MSE (3.21) (see Fig. 2d and Table II). A moderate R$^2$ of 0.38, followed closely by the mechanistic (MSE = 3.6, R$^2$ = 0.36) (see Fig. 2a) and Lasso models (3.72, 0.35), while XGBoost (4.71, 0.25), SVR (5.34, 0.25), and TabNet (4.24, 0.84) lagged considerably. In Cohort 2, the hybrid model retained its superiority (MSE = 2.01, R$^2$ = 0.58) (See Fig. 2d), followed by mechanistic (2.61, 0.54), Lasso (2.87, 0.52), XGBoost (3.83, 0.32), SVR (4.26, 0.26), and TabNet (4.01, 0.32) (see Table II). Although the R$^2$ values for [ΔNa$^+$] were lower than for [S-Na$^+$], this is attributable to the narrow physiological range of [ΔNa$^+$] (~±4 mEq/L) and the influence of measurement uncertainty (±1mEq/L), which amplifies residual variance and deflates R$^2$. Consequently, R$^2$ becomes less informative in this setting, even when absolute prediction errors remain clinically acceptable.

To overcome the interpretive limitations of R$^2$ for [ΔNa$^+$], we employed BER as a more robust and clinically interpretable metric, which considers the experimental uncertainty. In Cohort 1, the hybrid model demonstrated the lowest BER (47.63%) (see Fig. 2c), outperforming Lasso (50.84%), mechanistic (55.44%), SVR (54.23%), XGBoost (55.93%), and TabNet (58.12%). Under controlled correction in Cohort 2, the hybrid model exhibited a significantly reduced BER of 30.00% (p < 0.05, McNemar's test) (see Fig. 2c), outperforming mechanistic (54.36%), Lasso (36.66%), SVR (53.33%), XGBoost (63.33%), and TabNet (43.33%). These reductions demonstrate the hybrid model's ability to maintain predictions within clinically acceptable deviation limits across patient populations with varying risk profiles. Statistical testing using the Wilcoxon signed-rank test confirmed significant improvements in squared error performance for the hybrid model compared to TabNet, XGBoost, and SVR in Cohort 1 (p < 0.05), and against all models in Cohort 2 (p < 0.05). Similarly, the hybrid model achieved significantly lower BERs than all comparators across both cohorts (p < 0.05, McNemar's test), underscoring its superior accuracy and robustness.

**Table II**: Comparative Analysis of the hybrid model with the other ML models for Prediction of Post-Treatment [S-Na$^+$] and [ΔNa$^+$]

| Models | Prediction of post-treatment [S-Na$^+$] | | | | | | [ΔNa$^+$] | |
| | C1 | | | C2 | | | C1 | C2 |
| | MSE | BER (%) | R$^2$ | MSE | BER (%) | R$^2$ | R$^2$ | R$^2$ |
|---|---|---|---|---|---|---|---|---|
| XGB | 5.37 | 55.93 | 0.83 | 3.7 | 63.33 | 0.92 | 0.25 | 0.32 |
| SVR | 5.34 | 54.23 | 0.83 | 4.26 | 53.33 | 0.9 | 0.25 | 0.26 |
| TabNet | 4.24 | 58.12 | 0.84 | 4.06 | 43.33 | 0.91 | 0.29 | 0.32 |
| **Hybrid** | **2.8** | **47.63** | **0.87** | **1.31** | **30** | **0.97** | **0.38** | **0.58** |

C1-Cohort 1, C2- Cohort 2, BER (%)-Binary Error Rate, XGB-XGBoost

Together, these findings demonstrate that while R$^2$ offers a high-level view of variance explained, it can be misleading under constrained physiological ranges. In contrast, MSE and BER provide more consistent and clinically relevant performance indicators. The hybrid model consistently

outperforms mechanistic and ML models across all metrics, exhibiting strong generalization and precise modeling of [S-Na$^+$] dynamics. By integrating mechanistic domain knowledge with data-driven learning, the hybrid framework effectively addresses inter-individual variability and data sparsity, offering a scalable, physiologically grounded, and high-performing tool for [S-Na$^+$] prediction in both our primary and high-risk DDAVP-treated cohorts.

### B. Internal and External Validation with 21 Features: Prediction of Post-treatment [S-Na$^+$] and [ΔNa$^+$]

Using a reduced set of 21 features, both internal (primary data) (Table III) and external (MIMIC-IV) validations (Table IV) showed a modest decrease in performance compared to the original models with 65 features. Nonetheless, the hybrid model consistently outperformed the mechanistic and ML models across all cohorts and evaluation metrics. In internal validation for post-treatment [S-Na$^+$] prediction, the hybrid model achieved an MSE (±SD) of 3.4 (± 2.53) and an R$^2$ of 0.85 (± 6.53) in Cohort 1, compared to 2.8 and 0.87 with 65 features. In Cohort 2, it maintained strong results with MSE (±SD) = 1.8 (± 2.21) and R$^2$ (± SD) = 0.94 (± 4.38), slightly below the original values of 1.31 (Fig. 2b) and 0.97 (Fig. 2a). BER increased moderately from 47.63% (Fig. 2c) to 50.21 ± 9.54% (Table III) in Cohort 1 and from 30.00% (Fig. 2c) to 41.22 ± 7.81% (Table III) in Cohort 2. For [ΔNa$^+$] prediction, the hybrid model-maintained R$^2$ values of 0.37 for cohort 1 and 0.44 for cohort 2, demonstrating stable trend estimation despite the reduced feature set. External validation further confirmed the robustness of the hybrid model. It achieved R$^2$ of 0.85 ± 7.14 for cohort 1 and 0.92 ± 5.84 for cohort 2 while predicting [S-Na$^+$] (Table IV), with lower MSE and BER than other models. For example, its BER remained at 52.27 ± 8.87% for cohort 1 and 46.3 ± 5.44% for cohort 2, compared to over 58 ± 6.89% for models like TabNet. Although performance varied with fewer features, the hybrid model showed consistently lower error rates.

**Table III:** Results from internal validation using 21 features with mechanistic and alternative ML models for predicting Post-Treatment [S-Na+] and [ΔNa+].

| Models | Internal Validation with 21 Features Prediction of post-treatment [S-Na$^+$] | | | | | | [ΔNa$^+$] | |
| | C1 | | | C2 | | | C1 | C2 |
| | MSE | BER(%) | R$^2$ | MSE | BER(%) | R$^2$ | R$^2$ | R$^2$ |
|---|---|---|---|---|---|---|---|---|
| Math | 5.22 | 55.21 | 0.84 | 3.9 | 53.14 | 0.87 | 0.35 | 0.41 |
| Lasso | 5.68 | 56.44 | 0.83 | 3.61 | 53.11 | 0.88 | 0.36 | 0.43 |
| TabNet | 6.1 | 59.81 | 0.83 | 4.12 | 58.21 | 0.87 | 0.27 | 0.30 |
| **Hybrid** | **3.4** | **50.21** | **0.85** | **1.8** | **41.22** | **0.94** | **0.37** | **0.44** |

**Table IV:** Results of the external validation with 21 features, using the same trained models from Table III.

| Models | External Validation with 21 Features Prediction of post-treatment [S-Na$^+$] | | | | | | [ΔNa$^+$] | |
| | C1 | | | C2 | | | C1 | C2 |
| | MSE | BER(%) | R$^2$ | MSE | BER(%) | R$^2$ | R$^2$ | R$^2$ |
|---|---|---|---|---|---|---|---|---|
| Math | 6.36 | 58.4 | 0.84 | 3.9 | 55.14 | 0.87 | 0.30 | 0.4 |
| Lasso | 7.14 | 59.12 | 0.81 | 3.61 | 53.11 | 0.88 | 0.32 | 0.39 |
| TabNet | 7.2 | 61.54 | 0.80 | 4.12 | 58.21 | 0.87 | 0.24 | 0.28 |
| **Hybrid** | **4.2** | **52.27** | **0.85** | **2.6** | **46.3** | **0.92** | **0.33** | **0.43** |

### C. Model Interpretability Analysis (65 vs 21 features)

Fig. 3 shows the top 8 features out of 65, ranked by their mean absolute SHAP values, which represent each feature's average contribution to the hybrid model's [S-Na$^+$]

predictions. Figs. 3a and 3b present SHAP summary plots for Cohorts 1 and 2. Each dot represents a treatment event, colored by the actual feature value (red = high, blue = low), with SHAP values on the x-axis showing the direction and strength of each feature's effect.

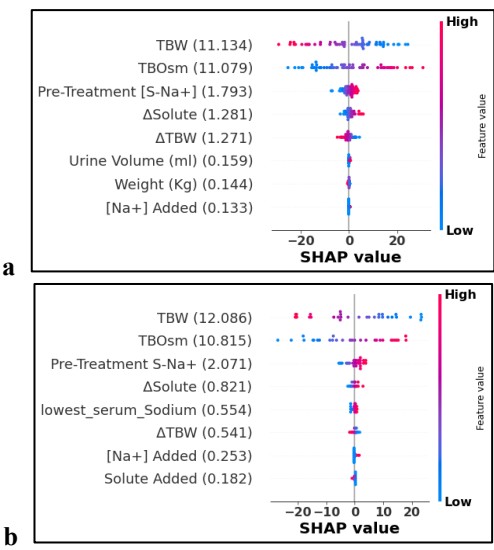

**Fig. 3**: SHAP Summary Plots Showing Feature Contributions in Hybrid Model for Unseen Data Prediction: a) SHAP plot for [S-Na$^+$] correction in Cohort 1, b) SHAP plot for [S-Na$^+$] in Cohort 2. The values in the bracket on y axis are the mean SHAP values for the corresponding feature. TBW-Total Body Water, TBOsm- Total Body Osmolality, ΔSolute-net interval solute, ΔTBW-Total Body Water Change, Solute Added- Solute added during treatment, [Na+] added- Sodium added from input Fluid, [U-K$^+$]-Urine Potassium, Urine [Osm]-Urine Osmolality Potassium Dose-Potassium added

Across both cohorts, five predictors, TBW, total body osmolality (TBOsm), Pre-Treatment [S-Na$^+$], ΔSolute, and ΔTBW, consistently emerged as key drivers of model performance, reflecting their physiological relevance and alignment with mechanistic model parameters. Additional variables such as body weight, solute input/output, and urine-based losses showed moderate contributions, enhancing prediction when integrated with core features.

Notably, features such as etiology and treatment duration, identified by Lasso as among the top contributors (Fig. S2), were excluded from the reduced 21-feature set due to their unavailability in external datasets, such as MIMIC-IV. This exclusion likely contributed to the observed decline in performance with fewer features, underscoring the additive value of clinically informative variables in model learning and predictive accuracy.

## V. DISCUSSION

This study demonstrates the technical utility of a hybrid modeling framework that integrates mechanistic equations with ML to predict [ΔNa$^+$] in hyponatremic patients, including those receiving DDAVP. Our results underscore the limitations of both standalone mechanistic and ML models, particularly when evaluated in real-world, data-constrained clinical settings. A key insight from our analysis is that the coefficient of determination ($R^2$) is an inadequate performance metric for predicting [ΔNa$^+$]. Although $R^2$ is commonly used to evaluate regression models, it quantifies the proportion of variance in the target variable explained by

the model, assuming homoscedasticity and a fixed error distribution. However, these assumptions often break down in clinical scenarios involving [ΔNa$^+$], where the observed variance in [ΔNa$^+$] is usually narrow and shaped by physiological constraints and guideline-based treatment ceilings (e.g., correction limits targeting 6 mEq/L/day). As a result, even small but clinically meaningful prediction errors can lead to deceptively low $R^2$ values, underrepresenting the model's practical utility and performance.

In such settings, $R^2$ can underrepresent model performance because it penalizes all deviations equally across the scale, regardless of their clinical impact. For instance, a prediction error of ±1 mEq/L while predicting [ΔNa$^+$] = 6 mEq/L may reduce $R^2$ by 18.8%. In contrast, the same error for predicting [S-Na$^+$] =125 mEq/L may reduce $R^2$ by only 2.7%, revealing a 7x greater penalty for ΔNa predictions despite identical error magnitudes. Yet, both are weighted identically in $R^2$ [31-33].

To overcome this discrepancy, we introduced BER as a clinically aligned performance metric that reflects whether the predicted correction falls within an acceptable clinical margin (±1 mEq/L) [34]. This framing is crucial when considering osmotic demyelination risk and other adverse outcomes, which are governed by clinical thresholds rather than statistical variance. The BER accounts for measurement uncertainty and the limited resolution of [S-Na$^+$] measurements (typically ±1 mEq/L) [4-5][34][35], making it a more appropriate and actionable metric in real-world decision-making.

Our hybrid model, trained on temporally enriched and physiologically guided features, consistently outperformed both mechanistic equation and standalone ML models, especially in the high-risk DDAVP subgroup. By incorporating temporal features and accounting for both treatment type and urine output trends, our model generated more stable and clinically plausible [ΔNa$^+$] predictions within a maximal 12-hour time period. In contrast, other studies only perform a 6-hour window.

Internal validation on the primary cohort, utilizing the complete set of 65 features, demonstrated superior overall performance. Notably, features such as etiology, treatment duration, and others were excluded from the reduced set due to their unavailability in MIMIC-IV and may have additive contributions to model learning. When retrained on the condensed 21-feature set, both internal and external validations exhibited some performance decline, likely attributable to the absence of these clinically significant variables. Despite this, the hybrid model consistently outperformed alternative models across both contexts. Models such as TabNet proved to be more sensitive to the reduced feature space, experiencing larger performance reductions. These findings suggest that, although comprehensive feature sets enhance accuracy, the hybrid model remains comparatively robust even with fewer variables and across different datasets.

A significant strength of this study is the completeness of our clinical datasets, which were precisely obtained by manual chart review from practicing clinical nephrologists. This strength contrasts with what we observed in other publicly available EHR-derived datasets, such as the MIMIC-

IV [21] dataset, where we found high levels of missingness (often exceeding 50% for key inputs, including urine electrolytes and fluid balances). ML models alone are vulnerable to such noise, but the hybrid strategy confers improved generalizability and robustness by incorporating current physiological models as well. This enables more accurate predictions, even when some clinical variables are missing or delayed, a common challenge in critical care settings.

Regarding temporal prediction windows, Cohort 1 includes both high-risk patients ($\leq$ 6-hour, DDAVP-treated) and lower-risk patients (>6-hour), leading to variable performance depending on the prediction interval. The highest accuracy was seen within the $\leq$6-hour window for high-risk patients, with an $R^2$ of 97.0 and a BER of 30%, supporting clinical use in urgent scenarios. Predictions up to 12 hours showed a decline ($R^2$ = 84.85 $\pm$ 4.21, BER = 51.22%), indicating a decrease in precision over longer periods. This trend reflects clinical practice where high-risk patients are monitored more frequently for timely treatment, while more stable patients can tolerate longer intervals. Although this study focused on static pointwise predictions due to the limited availability of longitudinal data, ongoing efforts aim to develop richer sequential datasets with multiple treatment events. These will enable the use of advanced sequential learning methods, such as transformers and continual learning, to improve personalized sodium trajectory predictions and enhance model accuracy and clinical value.

## VI. CONCLUSION AND FUTURE DIRECTIONS

This study addresses key limitations in [$\Delta Na^+$] modeling by combining a physiological mechanistic framework with machine learning (ML) techniques. The hybrid model integrates this refined equation with Lasso regression, utilizing approximately 65 clinical variables across a 12-hour prediction window. Feature importance analysis identified physiologically relevant variables such as TBW, TBOsm, $\Delta$Solute, and $\Delta$TBW as the main predictors. Incorporating fluid exchange data, demographic factors, and possible underlying causes further enhanced the model's predictive capacity. The better performance observed in high-risk Cohort 2 patients who were treated with DDAVP is expected, as DDAVP's role in physiologically concentrating urine reduces variability in hourly urine output.

The hybrid model achieved MSEs of 2.8 ($R^2$ = 0.87) for the primary cohort and 1.31 ($R^2$ = 0.97; p < 0.05) for the DDAVP subgroup. Recognizing $R^2$'s limitations, especially in narrow [$\Delta Na^+$] ranges where errors are sensitive, we developed a new Binary Error Rate (BER) metric that marks predictions differing by over $\pm$ 1 mEq/L as incorrect. The hybrid model significantly reduced the BER, from 54% with the mechanistic model to 30% with the hybrid in the DDAVP group (p < 0.05), indicating greater reliability. Ultimately, the hybrid approach outperformed purely mechanistic and ML-only models across all performance measures with statistical significance. These findings suggest that MSE and BER offer more dependable evaluation than $R^2$ for [$\Delta Na^+$] predictions across diverse [S- Na +] ranges and correction protocols.

Internal validation using all 65 features showed high predictive accuracy. However, retraining with a reduced set of 21 features, common to both the primary dataset and MIMIC—IV, decreased performance, likely because additional factors like etiology, active clinical issues, volume status, and medical history (see Table S1) contribute to predictions, though these variables were missing from MIMIC- IV. Nonetheless, the hybrid model consistently outperformed alternative methods in both internal and external tests, demonstrating robustness across different datasets and variable availability.

Although our model does not yet meet clinical implementation standards, this work marks significant progress toward models that better predict [$\Delta Na^+$] by merging physiological understanding with data-driven methods. While our study extended the prediction window to 12 hours, most predictions occurred within 6 hours, especially in the high-risk DDAVP group. Future research will focus on external validation using larger, more diverse cohorts, exploring transformer-based models for enhanced temporal accuracy, and conducting stratified analyses of DDAVP-treated patients across different time frames to improve the robustness of predictions.

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

# Supplement

Table S1: The details of the 65 clinical Features Used in this study

| Feature | Description | Feature | Description |
|---|---|---|---|
| --- **Serum, Urine, and Calculated Metrics** --- | | | |
| Pre-Interval Na+ | Serum sodium at start (mEq/L) | Lowest Interval Na+ | Lowest serum sodium during interval (mEq/L) |
| ΔSolute | Total body (Na+K) change / 2 (neg. charges) | Solute Added | [Na+K] added during treatment |
| [Na+] added | Sodium added during treatment | Potassium Osm | Osmoles from potassium |
| TBOsm | Total body osmoles before treatment | TBW | Total body water before treatment |
| ΔTBW | Change in total body water | Urine Volume (ml) | Total urine output (mL) |
| [U-Na] | Urine sodium concentration (mEq/L) | [U-K] | Urine potassium concentration (mEq/L) |
| [U-Osm] | Urine osmolality (mOsm/kg) | | |
| --- **Treatment Duration & Demographics** --- | | | |
| Treatment Duration (Hours) | Duration of treatment interval | Age | Patient age in years |
| Gender | Patient gender | Height | Height in cm |
| Weight (Kg) | Patient weight in kilograms | | |
| --- **Infused & Oral Fluids** --- | | | |
| Normal Saline (0.9%) | Volume of 0.9% NaCl given (mL) | Half Normal Saline | Volume of 0.45% NaCl given (mL) |
| D5W | Volume of 5% dextrose water given (mL) | Hypertonic Saline (3%) | Volume of 3% NaCl given (mL) |
| Lactate Ringer | Volume of LR solution given (mL) | PO | Oral intake during treatment (mL) |
| Blood_in | Blood transfusion volume (mL) | Unspecified_Fluid_In | Other IV fluids not specified (mL) |
| --- **Manually Logged Inputs/Outputs** --- | | | |
| Manual_Water_In | Manually logged water intake (mL) | Manual_Solute_In | Manually logged solute intake (mmol) |
| other_out | Other outputs (e.g., drains, tubes) | unmeasured_stools | Estimated stool fluid loss (mL) |
| --- **Medications & Interventions** --- | | | |
| Salt_Tabs_Dose | Salt tablet dose given (g) | Potassium_Dose | Potassium dose given (mmol) |
| Loop_Dose | Loop diuretic dose (mg) | Medication_Thiazide | On thiazide diuretics (Yes/No) |
| DDAVP | Desmopressin use (Yes/No) | Pressors | On vasopressors (Yes/No) |
| Inotropes | On inotropes (Yes/No) | | |
| --- **Acute State & Monitoring** --- | | | |
| Intubated | Patient intubated (Yes/No) | Foley | Foley catheter present (Yes/No) |
| Fever | Fever present (Yes/No) | | |
| --- **Etiology of Hyponatremia** --- | | | |
| Etiology_Hypovolemic | Due to hypovolemia (Yes/No) | Etiology_SIADH | Due to SIADH (Yes/No) |
| Etiology_Polydipsia | Due to excess water intake (Yes/No) | Etiology_Low Solute Intake | Due to low solute diet (Yes/No) |
| Etiology_Other | Other causes (Yes/No) | | |
| --- **Active Clinical Problems** --- | | | |
| Active_Probs_CHF | Current CHF episode (Yes/No) | Active_Probs_Liver | Active liver disease (Yes/No) |
| Active_Probs_Seizure | Current seizure activity (Yes/No) | Active_Probs_AMS | Altered mental status (Yes/No) |
| Active_Probs_Vomiting | Active vomiting (Yes/No) | Active_Probs_Diarrhea | Active diarrhea (Yes/No) |
| --- **Past Medical History** --- | | | |
| Medical History_Cirrhosis | History of liver cirrhosis (Yes/No) | Medical History: CHF | History of CHF (Yes/No) |
| Medical History_Hyponatremia | Prior hyponatremia episodes (Yes/No) | Medical History_Alcohol Use Disorder | Alcohol use disorder history (Yes/No) |
| Medical History_Bipolar | Bipolar disorder (Yes/No) | Medical History_Schizophrenia | Schizophrenia history (Yes/No) |
| Medical History_Dementia | Dementia history (Yes/No) | Medical History_Depression | Depression history (Yes/No) |
| Medical History_Malignancy | Cancer history (Yes/No) | Medical History_Metastatic Cancer | Metastatic cancer history (Yes/No) |
| --- **Volume Status Assessment** --- | | | |
| Volume Status_Euvolemia | Clinically euvolemic (Yes/No) | Volume Status_Hypervolemic | Clinically hypervolemic (Yes/No) |
| Volume Status_Hypovolemic | Clinically hypovolemic (Yes/No) | Volume Status_Unknown | Volume status unclear (Yes/No) |

*Inclusion and Exclusion criteria for MIMIC-IV (External Validation*

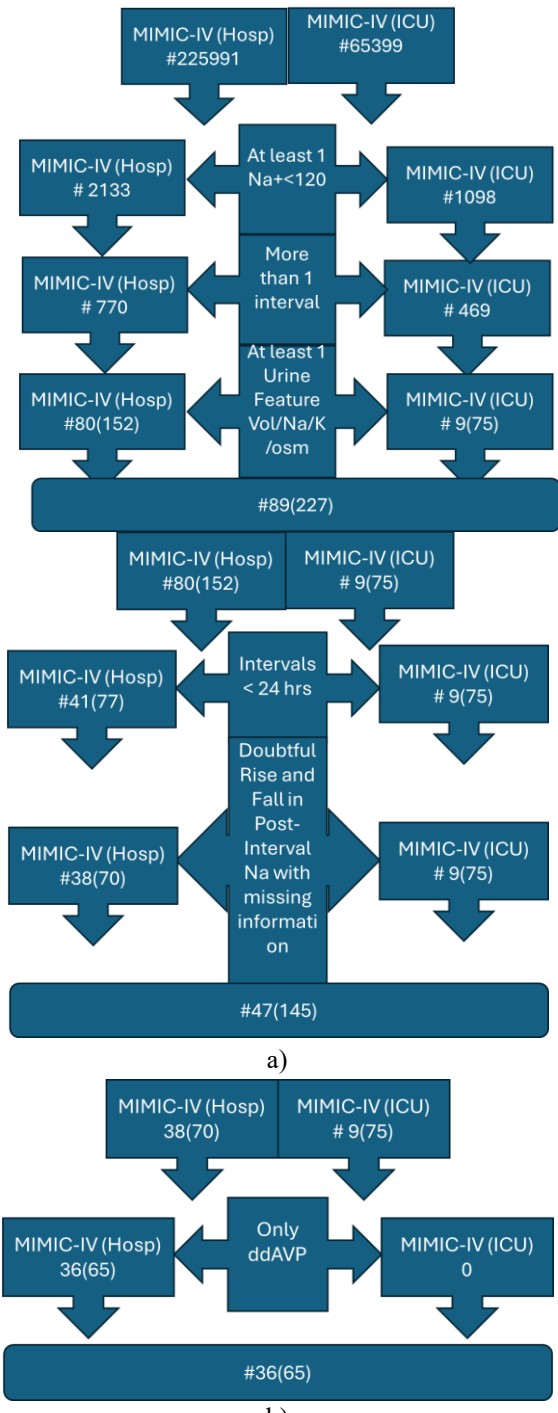

a)

b)

Figure S1: Inclusion and exclusion criteria for MIMIC-IV to identify severe hyponatremia patients: a) Cohort 1, and b) Cohort 2

Table S2: List of common features from internal (Table S1) and external validation data

| Features in internal Validation data | MIMIC-IV | Features in internal Validation data | MIMIC-IV |
|---|---|---|---|
| Sodium Correction (ΔNa) | 2 | Gender | 1 |
| Post Interval Na+ | 1 | Height | 1 |
| Pre-Interval Na+ | 1 | Normal Saline (0.9% NaCl) | 1 |
| Interval Duration (Hours) | 1 | Half Normal Saline (0.45% NaCl) | 1 |
| ΔTBW | 2 | D5W | 1 |
| ΔSolute | 2 | Hypertonic Saline (3% NaCl) | 1 |
| TBW | 2 | Lactate Ringer | 1 |
| TBOsm | 2 | Potassium Dose | 1 |
| Weight (Kg) | 1 | | |
| Urine Volume (ml) | 1 | | |
| Urine [K+] | 0/1 | | |
| Urine [Na] | 0/1 | | |
| Urine [Osm] | 0/1 | | |
| Foley | 1/1 | | |
| Age | 1 | | |

0- Not Given/No information, 1- Present, 2- need to calculate / indirect approach

*Model Interpretability for Lasso:*

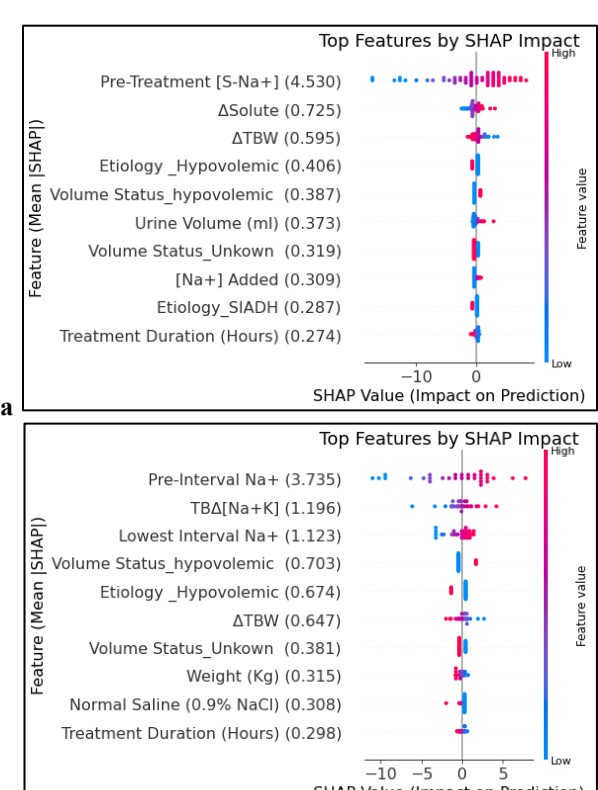

**Fig. S2**: SHAP Summary Plots Showing Feature Contributions in Lasso Model for Unseen Data Prediction with 65 features: a) SHAP plot for [S-Na+] correction in Cohort 1, b) SHAP plot for [S-Na+] in Cohort 2. The values in the bracket on y axis are the mean SHAP values for the corresponding feature. ΔSolute-net interval solute, ΔTBW-Total Body Water Change, [Na+] added- Sodium added from input Fluid