# OpenReview forum: "A Hybrid Learning Framework for Predicting Post-Treatment Serum Sodium in Patients with Hyponatremia"
_IEEE.org/EMBS/BHI/2025/Conference — BHI 2025_

### Official Review · Reviewer_2L8e · 2025-06-30
**A Hybrid Learning Framework for Predicting Post-Treatment Serum Sodium in Patients with Hyponatremia**

**Confidence:** 4
**Clarity Of Writing:** great
**Clinical Significance:** great
**Methodological Novelty:** great
**Overall Rating:** 8

**Experiments And Results:**

great

**Questions For The Authors:**

1. The machine learning component employs standard algorithms such as Lasso, Support Vector Regression (SVR), and XGBoost, rather than more advanced or contemporary architectures like deep learning models or transformer-based networks. Clarification is needed regarding the rationale for this selection, particularly given that the chosen methods may have limited capacity to model temporal dependencies and uncertainty inherent in clinical data.

2. The model predicts sodium correction over a 12-hour time window, which—although longer than the horizons considered in some previous studies—may still be insufficient for comprehensive, long-term treatment planning. The basis for selecting this prediction window should be explicitly justified.

3. The model was trained and validated using data from only two institutions (Massachusetts General Hospital and Brigham and Women’s Hospital). This may raise concerns about the model’s generalizability to other healthcare settings or diverse patient populations. A discussion on potential biases in the dataset would be beneficial.

4. The inclusion of case studies or visual tools, such as dashboards that incorporate prediction results and SHAP value interpretations, could enhance the practical relevance of the work and illustrate how the model might support clinical decision-making.

5. The manuscript would be strengthened by a high-level roadmap outlining the integration of the proposed model into electronic health record (EHR) systems. This should include considerations related to data flow, alert thresholds, clinician interface, and workflow integration.

6. The paper does not benchmark its hybrid model against modern deep learning approaches, which are increasingly prevalent in clinical prediction tasks. Including such a comparison would provide important context and potentially validate the advantages of the proposed methodology.

7. The dataset comprises a relatively small cohort (144 patients), yet the study utilizes complex models such as XGBoost and SVR. In the absence of external validation, there is a considerable risk of overfitting, especially given the high dimensionality of the feature space. This limitation should be addressed, and additional validation efforts are recommended.

**Strengths:**

1. High quality data and rich features
2. The use of SHAP values allows for transparent understanding of which clinical features most influence predictions, aiding clinical trust and adoption.

3. Combining mechanistic and data-driven approaches, the model is more resilient to missing data and better suited for diverse clinical settings

4. Hybrid model outperforms both traditional physiological equations and standalone machine learning models in predicting post-treatment serum sodium levels and sodium correction (ΔNa⁺), especially in high-risk patients.

**Summary Of The Paper:**

Authors have presented a hybrid learning framework that combines physiology with machine learning to predict post-treatment serum sodium levels in patients with severe hyponatremia. By integrating clinical data and mechanistic models, the approach improves prediction accuracy, especially in high-risk patients treated with desmopressin (DDAVP). The model outperforms traditional methods and introduces a new metric, Binary Error Rate (BER), to better reflect clinical relevance.

**Weaknesses:**

1. Although the model extends prediction to a 12-hour window (longer than some prior studies), it still may not capture longer-term trends or delayed treatment effects

2.Despite improved robustness, the model still relies on a relatively complete set of clinical features (65 variables), which may not always be available in real-time clinical environments.

3.The study uses traditional ML models (Lasso, SVR, XGBoost) and does not yet explore more advanced temporal models like transformers, which could better capture time-series dynamics

---

### Official Review · Reviewer_Uu49 · 2025-07-17
**Great writing while combined ML models seems to be redundant**

**Confidence:** 3
**Clarity Of Writing:** great
**Clinical Significance:** good
**Methodological Novelty:** poor
**Overall Rating:** 4
**Final Rating:** 6

**Experiments And Results:**

good

**Questions For The Authors:**

First, given that Lasso seems to always outperform in the ML models and in your framework ML output come from the best performing model, is it fair to say that ensemble of Lasso and mechanical model will give us similar results? If so, why we need to consider combinations of ML models? Second, if SHAP provide informative messages on the feature contributions, will showing SHAP results in mechanical model and ML models separately and comparing it with the ensemble SHAP help illustrate why hybrid model could achieve better performance with respect to feature contribution difference? Third, did you evaluate the time-to-prediction vs. outcome accuracy?
For instance, would a 6-hour prediction window yield higher BER or lower MSE than 12-hour? A comparison would inform how frequently the model should be queried in practice.

**Strengths:**

The integration of mechanistic modeling with ML is a sound and novel strategy that addresses common shortcomings in both paradigms—poor generalizability in ML, and limited adaptability in mechanistic models. The use of manually reviewed, high-resolution chart data with precise interval curation (<12 hr) provides a rare and high-fidelity dataset. The exclusion of physiologically impossible cases enhances data reliability. The model targets a critical and complex clinical decision—safe correction of severe hyponatremia—where prediction accuracy can reduce serious complications like osmotic demyelination syndrome. The introduction of Binary Error Rate acknowledges limitations of R² in tightly constrained physiological domains and helps bridge the gap between statistical metrics and clinical decision-making. SHAP analysis was used to interpret the contribution of features, which helps clinicians understand and potentially trust the model outputs.

**Summary Of The Paper:**

This paper presents a hybrid predictive framework that combines mechanistic physiological modeling with machine learning (ML) to predict post-treatment serum sodium concentrations and the correction in hospitalized patients with hyponatremia, particularly those treated with 3% hypertonic saline and DDAVP (desmopressin). The model was trained and validated on two manually curated clinical datasets from Massachusetts General Hospital and Brigham and Women’s Hospital, comprising:Cohort 1 (n=144): Mixed-risk hyponatremic ICU patients. Cohort 2 (n=73): High-risk DDAVP-treated subset.

The hybrid model integrates a modified Rose equation for mechanistic estimation, a machine learning ensemble of Lasso regression, SVR, and XGBoost and a late fusion strategy combining mechanistic and ML predictions equally. It introduces a new evaluation metric, Binary Error Rate (BER), to assess clinical reliability within ±1 mEq/L, reflecting the clinical significance of small prediction errors. The model consistently outperforms both standalone physiological equations and ML models on mean squared error (MSE), R², and BER—particularly in high-risk patients.

**Weaknesses:**

While model performance is impressive, the paper lacks discussion on how the model would be integrated into real-time workflows (e.g., via EHR decision support) and how often predictions would be updated as new data come in. Besides, only classical equations and basic ML regressors are used, which limits the competitiveness of comparisons.

---

### Official Review · Reviewer_AZxr · 2025-07-17
**A Hybrid Learning Framework for Predicting Post-Treatment Serum Sodium in Patients with Hyponatremia**

**Confidence:** 3
**Clarity Of Writing:** good
**Clinical Significance:** great
**Methodological Novelty:** good
**Overall Rating:** 6

**Experiments And Results:**

good

**Questions For The Authors:**

This is a strong paper with high clinical impact potential. With minor improvements and external validation, it could serve as a valuable tool for managing hyponatremia safely and effectively.

**Strengths:**

1) The hybrid framework is novel and effective, improving prediction accuracy over baseline models.

2) Evaluation using clinically meaningful metrics (±1 mEq/L tolerance, BER) is appropriate and insightful.

3) Focus on DDAVP-treated patients strengthens the clinical relevance.

4) Use of high-quality, curated data and robust validation methods enhances credibility.

**Summary Of The Paper:**

This paper addresses a clinically important problem with a well-designed hybrid model combining physiological equations and machine learning. The work is clearly written, methodologically sound, and offers strong potential for clinical utility, especially in high-risk DDAVP-treated patients.

**Weaknesses:**

1) Broader Baselines: Include comparisons with other physiological models like Edelman’s.

2) Model Deployment: Discuss reduced-variable versions for practical clinical use.
3)Justify ±1 mEq/L threshold or provide sensitivity analysis

---

### Official Review · Reviewer_qMsT · 2025-07-17
**A solid hybrid approach with detailed methodology, but lacks external validation and justification for its new evaluation metric**

**Confidence:** 4
**Clarity Of Writing:** good
**Clinical Significance:** great
**Methodological Novelty:** good
**Overall Rating:** 3
**Final Rating:** 6

**Experiments And Results:**

poor

**Questions For The Authors:**

My main concern regarding the paper includes the lack of external validation and baseline comparison, and the lack of justification for the BER metric.

**Strengths:**

- The paper presents a detailed and well-structured methodology for developing a hybrid model that combines mechanistic and machine learning approaches.

**Summary Of The Paper:**

Monitoring Post-treatment serum sodium levels in patients is crucial to prevent complications associated with hyponatremia (low-sodium).
While mechanistic and machine learning (ML) models both offered valuable insights, they were only evaluated on non-individualized predictions on small and homogenous datasets, thus limiting generalizability.
The paper attempts to address the gap by developing a hybrid approach that combines mechanistic and ML models to predict post-treatment serum sodium levels. In addition to the hybrid model, the paper also contributes a high-quality cohort curated by nephrologists, a new evaluation metric called binary error rate (BER), and a dedicated evaluation on a high-risk subset.

**Weaknesses:**

- While the paper mentions "a prediction error of ±1 mEq/L near a correction threshold of 6 mEq/L may influence a critical treatment decision, whereas the same error at a relatively safer value > 125 mEq/L may not be as clinically relevant", it does not provide sufficient justification for the choice of introducing BER by measuring the percentage of predictions with less than 1mEq/L error, which is not a standard evaluation and lacks clarity on its significance. References to existing literature or clinical guidelines that support this choice would strengthen the argument.
- Additionally, it's unclear whether BER can be applied to other clinical prediction tasks (i.e. hypo/hyperkalemia), which raises concerns about its generalizability and applicability in broader contexts.
- The paper claims, "a significant strength of this study is the completeness of our clinical datasets, which were meticulously obtained by manual chart review from practicing clinical nephrologists." While data completeness and quality are indeed vital, it fails to translate this paper's method into a clear advantage over existing datasets or methods. Even if the initial modeling and data curation was developed on a closed dataset, the paper could still conduct external validation on the same open, larger, more sparse datasets used by other prior studies to validate the advantage of their approach.
- Given the recent popularity of deep neural networks, the paper is encouraged to explore the potential of deep learning models in conjunction with their hybrid approach, or discuss why they were not considered. Given the hybrid model could easily be extendned to incorporate deep learning models instead of traditional ML models, it would be beneficial to discuss the potential advantages or limitations of such an approach.

---

### Official Review · Reviewer_MWVK · 2025-07-17

[review text omitted: it was posted to a different submission]